# Simple Strategies for Recovering Inner Products from Coarsely Quantized Random Projections

**Ping Li**
Baidu Research, and
Rutgers University
pingli98@gmail.com

**Martin Slawski**
Department of Statistics
George Mason University
mslawsk3@gmu.edu

## Abstract

Random projections have been increasingly adopted for a diverse set of tasks in machine learning involving dimensionality reduction. One specific line of research on this topic has investigated the use of quantization subsequent to projection with the aim of additional data compression. Motivated by applications in nearest neighbor search and linear learning, we revisit the problem of recovering inner products (respectively cosine similarities) in such setting. We show that even under coarse scalar quantization with 3 to 5 bits per projection, the loss in accuracy tends to range from "negligible" to "moderate". One implication is that in most scenarios of practical interest, there is no need for a sophisticated recovery approach like maximum likelihood estimation as considered in previous work on the subject. What we propose herein also yields considerable improvements in terms of accuracy over the Hamming distance-based approach in Li et al. (ICML 2014) which is comparable in terms of simplicity.

## 1 Introduction

The method of random projections (RPs) for linear dimensionality reduction has become more and more popular over the years after the basic theoretical foundation, the celebrated Johnson-Lindenstrauss (JL) Lemma [12, 20, 33], had been laid out. In a nutshell, it states that it is possible to considerably lower the dimension of a set of data points by means of a linear map in such a way that squared Euclidean distances and inner products are roughly preserved in the low-dimensional representation. Conveniently, a linear map of this sort can be realized by a variety of random matrices [1, 2, 18]. The scope of applications of RPs has expanded dramatically in the course of time, and includes dimension reduction in linear classification and regression [14, 30], similarity search [5, 17], compressed sensing [8], clustering [7, 11], randomized numerical linear algebra and matrix sketching [29], and differential privacy [21], among others.

The idea of achieving further data compression by means of subsequent scalar quantization of the projected data has been considered for a while. Such setting can be motivated from constraints concerning data storage and communication, locality-sensitive hashing [13, 27], or the enhancement of privacy [31]. The extreme case of one-bit quantization can be associated with two seminal works in computer science, the SDP relaxation of the MAXCUT problem [16] and the simhash [10]. One-bit compressed sensing is introduced in [6], and along with its numerous extensions, has meanwhile developed into a subfield within the compressed sensing literature. A series of recent papers discuss quantized RPs with a focus on similarity estimation and search. The papers [25, 32] discuss quantized RPs with a focus on image retrieval based on nearest neighbor search. Independent of the specific application, [25, 32] provide JL-type statements for quantized RPs, and consider the trade-off between the number of projections and the number of bits per projection under a given budget of bits as it also appears in the compressed sensing literature [24]. The paper [19] studies approximate JL-type results for quantized RPs in detail. The approach to quantized RPs taken in the present paper follows [27, 28]

in which the problem of recovering distances and inner products is recast within the framework of classical statistical point estimation theory. The paper [28] discusses maximum likelihood estimation in this context, with an emphasis on the aforementioned trade-off between the number of RPs and the bit depth per projection. In the present paper we focus on the much simpler and computationally much more convenient approach in which the presence of the quantizer is ignored, i.e., quantized data are treated in the same way as full-precision data. We herein quantify the loss of accuracy of this approach relative to the full-precision case, which turns out to be insignificant in many scenarios of practical interest even under coarse quantization with 3 to 5 bits per projection. Moreover, we show that the approach compares favorably to the Hamming distance-based (or equivalently collision-based) scheme in [27] which is of similar simplicity. We argue that both approaches have their merits: the collision-based scheme performs better in preserving local geometry (the distances of nearby points), whereas the one studied in more detail herein yields better preservation globally.

**Notation.** For a positive integer $m$, we let $[m] = \{1, \ldots, m\}$. For $l \in [m]$, $v_{(l)}$ denotes the $l$-th component of a vector $v \in \mathbb{R}^m$; if there is no danger of confusion with another index, the brackets in the subscript are omitted. $I(P)$ denotes the indicator function of expression $P$.

**Supplement**: Proofs and additional experimental results can be found in the supplement.

**Basic setup.** Let $\mathcal{X} = \{x_1, \ldots, x_n\} \subset \mathbb{R}^d$ be a set of input data with squared Euclidean norms $\lambda_i^2 := \|x_i\|_2^2$, $i \in [n]$. We think of $d$ being large. RPs reduce the dimensionality of the input data by means of a linear map $A : \mathbb{R}^d \to \mathbb{R}^k$, $k \ll d$. We assume throughout the paper that the map $A$ is realized by a random matrix with i.i.d. entries from the standard Gaussian distribution, i.e., $A_{lj} \sim N(0, 1)$, $l \in [k]$, $j \in [d]$. One standard goal of RPs is to approximately preserve distances in $\mathcal{X}$ while lowering the dimension, i.e., $\|Ax_i - Ax_j\|_2^2 / k \approx \|x_i - x_j\|_2^2$ for all $(i, j)$. This is implied by approximate inner product preservation $\langle x_i, x_j \rangle \approx \langle Ax_i, Ax_j \rangle / k$ for all $(i, j)$.

For the time being, we assume that it is possible to compute and store the squared norms $\{\lambda_i^2\}_{i=1}^n$, and to rescale the input data to unit norm, i.e., one first forms $\widetilde{x}_i \leftarrow x_i / \lambda_i$, $i \in [n]$, before applying $A$. In this case, it suffices to recover the (cosine) similarities $\rho_{ij} := \frac{\langle x_i, x_j \rangle}{\lambda_i \lambda_j} = \langle \widetilde{x}_i, \widetilde{x}_j \rangle$, $i, j \in [n]$, of the input data $\mathcal{X}$ from their compressed representation $\mathcal{Z} = \{z_1, \ldots, z_n\}$, $z_i := A\widetilde{x}_i$, $i \in [n]$.

## 2 Estimation of cosine similarity based on full-precision RPs

As preparation for later sections, we start by providing background concerning the usual setting without quantization. Let $(Z, Z')_r$ be random variables having a bivariate Gaussian distribution with zero mean, unit variance, and correlation $r \in (-1, 1)$:

$$(Z, Z')_r \sim N_2 \left( \begin{pmatrix} 0 \\ 0 \end{pmatrix}, \begin{pmatrix} 1 & r \\ r & 1 \end{pmatrix} \right). \tag{1}$$

Let further $x, x'$ be a generic pair of points from $\mathcal{X}$, and let $z := A\widetilde{x}$, $z' := A\widetilde{x}'$ be the counterpart in $\mathcal{Z}$. Then the components $\{(z_{(l)}, z'_{(l)})\}_{l=1}^k$ of $(z, z')$ are distributed i.i.d. as in (1) with $r = \rho =: \langle \widetilde{x}, \widetilde{x}' \rangle$. Hence the problem of recovering the cosine similarity of $x$ and $x'$ can be re-cast as estimating the correlation from an i.i.d. sample of $k$ bivariate Gaussian random variables. To simplify our exposition, we henceforth assume that $0 \le \rho < 1$ as this can easily be achieved by flipping the sign of one of $x$ or $x'$. The standard estimator of $\rho$ is what is called the "linear estimator" herein:

$$\widehat{\rho}_{\text{lin}} = \frac{1}{k} \langle z, z' \rangle = \frac{1}{k} \sum_{l=1}^k z_{(l)} z'_{(l)}. \tag{2}$$

As pointed out in [26] this estimator can be considerably improved upon by the maximum likelihood estimator (MLE) given (1):

$$\widehat{\rho}_{\text{MLE}} = \underset{r}{\arg\max} \left\{ -\frac{1}{2} \log(1 - r^2) - \frac{1}{2} \frac{1}{1 - r^2} \left( \frac{1}{k} \|z\|_2^2 + \frac{1}{k} \|z'\|_2^2 - \frac{1}{k} \langle z, z' \rangle 2r \right) \right\}. \tag{3}$$

The estimator $\widehat{\rho}_{\text{MLE}}$ is not available in closed form, which is potentially a serious concern since it needs to be evaluated for numerous different pairs of data points. However, this can be addressed

by tabulation of the two statistics $\left\{ \left( \|z\|_2^2 + \|z'\|_2^2 \right)/k, \, \langle z, z' \rangle /k \right\}$ and the corresponding solutions $\widehat{\rho}_{\mathrm{MLE}}$ over a sufficiently fine grid. At processing time, computation of $\widehat{\rho}_{\mathrm{MLE}}$ can then be reduced to a look-up in a pre-computed table.

One obvious issue of $\widehat{\rho}_{\mathrm{lin}}$ is that it does not respect the range of the underlying parameter. A natural fix is the use of the "normalized linear estimator"

$$\widehat{\rho}_{\mathrm{norm}} = \langle z, z' \rangle /(\|z\|_2 \, \|z'\|_2). \tag{4}$$

When comparing different estimators of $\rho$ in terms of statistical accuracy, we evaluate the mean squared error (MSE), possibly asymptotically as the number of RPs $k \to \infty$. Specifically, we consider

$$\mathrm{MSE}_\rho(\widehat{\rho}) = \mathbf{E}_\rho[(\rho - \widehat{\rho})^2] = \mathrm{Bias}_\rho^2(\widehat{\rho}) + \mathrm{Var}_\rho(\widehat{\rho}), \qquad \mathrm{Bias}_\rho(\widehat{\rho}) := \mathbf{E}_\rho[\widehat{\rho}] - \rho, \tag{5}$$

where $\widehat{\rho}$ is some estimator, and the subscript $\rho$ indicates that expectations are taken with respect to a sample $(z, z')$ following the bivariate normal distribution in (1) with $r = \rho$.

It turns out that $\widehat{\rho}_{\mathrm{norm}}$ and $\widehat{\rho}_{\mathrm{MLE}}$ can have dramatically lower (asymptotic) MSEs than $\widehat{\rho}_{\mathrm{lin}}$ for large values of $\rho$, i.e., for points of high cosine similarity. It can be shown that (cf. [4], p.132, and [26])

$$\mathrm{Bias}_\rho(\widehat{\rho}_{\mathrm{lin}}) = 0, \qquad\qquad \mathrm{Var}_\rho(\widehat{\rho}_{\mathrm{lin}}) = (1 + \rho^2)/k, \tag{6}$$

$$\mathrm{Bias}_\rho^2(\widehat{\rho}_{\mathrm{norm}}) = O(1/k^2), \quad \mathrm{Var}_\rho(\widehat{\rho}_{\mathrm{norm}}) = (1 - \rho^2)^2/k + O(1/k^2), \tag{7}$$

$$\mathrm{Bias}_\rho^2(\widehat{\rho}_{\mathrm{MLE}}) = O(1/k^2), \quad \mathrm{Var}_\rho(\widehat{\rho}_{\mathrm{MLE}}) = \frac{(1-\rho^2)^2}{1+\rho^2}/k + O(1/k^2). \tag{8}$$

While for $\rho = 0$, the (asymptotic) MSEs are the same, we note that the leading terms of the MSEs of $\widehat{\rho}_{\mathrm{norm}}$ and $\widehat{\rho}_{\mathrm{MLE}}$ decay at rate $\Theta((1 - \rho)^2)$ as $\rho \to 1$, whereas the MSE of $\widehat{\rho}_{\mathrm{lin}}$ grows with $\rho$. The following table provides the asymptotic MSE ratios of $\widehat{\rho}_{\mathrm{lin}}$ and $\widehat{\rho}_{\mathrm{norm}}$ for selected values of $\rho$.

| $\rho$ | 0.5 | 0.6 | 0.7 | 0.8 | 0.9 | 0.95 | 0.99 |
|---|---|---|---|---|---|---|---|
| $\frac{\mathrm{MSE}_\rho(\widehat{\rho}_{\mathrm{lin}})}{\mathrm{MSE}_\rho(\widehat{\rho}_{\mathrm{norm}})}$ | 2.2 | 3.3 | 5.7 | 12.6 | 50 | 200 | 5000 |

In conclusion, if it is possible to pre-compute and store the norms of the data prior to dimensionality reduction, a simple form of normalization can yield important benefits with regard to the recovery of inner products and distances for pairs of points having high cosine similarity. The MLE can provide a further refinement, but the improvement over $\widehat{\rho}_{\mathrm{norm}}$ can be at most by a factor of 2.

## 3 Estimation of cosine similarity based on quantized RPs

The following section contains our main results. After introducing preliminaries regarding quantization, we review previous approaches to the problem, before analyzing estimators following a different paradigm. We conclude with a comparison and some recommendations about what to use in practice.

**Quantization.** After obtaining the projected data $\mathcal{Z}$, the next step is scalar quantization. Let $\mathbf{t} = (t_1, \ldots, t_{K-1})$ with $0 = t_0 < t_1 < \ldots < t_{K-1} < t_K = +\infty$ be a set of *thresholds* inducing a partitioning of the positive real line into $K$ intervals $\{[t_{s-1}, t_s), \, s \in [K]\}$, and let $\mathcal{M} = \{\mu_1, \ldots, \mu_K\}$ be a set of *codes* with $\mu_s$ representing interval $[t_{s-1}, t_s), \, s \in [K]$. Given $\mathbf{t}$ and $\mathcal{M}$, the scalar quantizer (or quantization map) is defined by

$$Q : \mathbb{R} \to \mathcal{M}^\pm := -\mathcal{M} \cup \mathcal{M}, \quad z \mapsto Q(z) = \mathrm{sign}(z) \sum_{s=1}^{K} \mu_s I(|z| \in [t_{s-1}, t_s)). \tag{9}$$

The projected and quantized data result as $\mathcal{Q} = \{q_i\}_{i=1}^n \subset (\mathcal{M}^\pm)^k$, $q_i = \left( Q(z_{i_{(l)}}) \right)_{l=1}^k$, where $z_{i_{(l)}}$ denotes the $l$-th component of $z_i \in \mathcal{Z}$, $l \in [k]$, $i \in [n]$. The bit depth $b$ of the quantizer is given by $b := 1 + \log_2(K)$. For simplicity, we only consider the case where $b$ is an integer. The case $b = 1$ is well-studied [10, 27] and is hence disregarded in our analysis to keep our exposition compact.

**Bin-based vs. code-based approaches.** Let $q = Q(z)$ and $q' = Q(z')$ be the points resulting from quantization of the generic pair $z, z'$ in the previous section. In this paper, we distinguish between two basic paradigms for estimating the cosine similarity of the underlying pair $x, x'$ from $q, q'$. The first paradigm, which we refer to as *bin-based* estimation, does not make use of the specific values of

the codes $\mathcal{M}^{\pm}$, but only of the intervals ("bins") associated with each code. This is opposite to the second paradigm, referred to as *code-based* estimation which only makes use of the values of the codes. As we elaborate below, an advantage of the bin-based approach is that working with intervals reflects the process of quantization more faithfully and hence can be statistically more accurate; on the other hand, a code-based approach tends to be more convenient from the point of view computation. In this paper, we make a case for the code-based approach by showing that the loss in statistical accuracy can be fairly minor in several regimes of practical interest.

**Lloyd-Max (LM) quantizer.** With $b$ respectively $K$ being fixed, one needs to choose the thresholds $\mathbf{t}$ and the codes $\mathcal{M}$ of the quantizer (the second is crucial only for a code-based approach). In our setting, with $z_{i(l)} \sim N(0,1)$, $i \in [n]$, $l \in [k]$, which is inherited from the distribution of the entries of $A$, a standard choice is LM quantization [15] which minimizes the squared distortion error:

$$(\mathbf{t}^\star, \boldsymbol{\mu}^\star) = \underset{\mathbf{t}, \boldsymbol{\mu}}{\operatorname{argmin}} \, \mathbf{E}_{g \sim N(0,1)}[\{g - Q(g; \mathbf{t}, \boldsymbol{\mu})\}^2]. \tag{10}$$

Problem (10) can be solved by an iterative scheme that alternates between optimization of $\mathbf{t}$ for fixed $\boldsymbol{\mu}$ and vice versa. That scheme can be shown to deliver the global optimum [22]. In the absence of any prior information about the cosine similarities that we would like to recover, (10) appears as a reasonable default whose use for bin-based estimation has been justified in [28]. In the limit of cosine similarity $\rho \to 1$, it may seem more plausible to use (10) with $g$ replaced by its square, and taking the root of the resulting optimal thresholds and codes. However, it turns out that empirically this yields reduced performance more often than improvements, hence we stick to (10) in the sequel.

### 3.1 Bin-based approaches

**MLE.** Given a pair $q = (q_{(l)})_{l=1}^k$ and $q' = (q'_{(l)})_{l=1}^k$ of projected and quantized points, maximum likelihood estimation of the underlying cosine similarity $\rho$ is studied in depth in [28]. The associated likelihood function $L(r)$ is based on bivariate normal probabilities of the form $\mathbf{P}_r(Z \in [t_{s-1}, t_s), Z' \in [t_{u-1}, t_u))$, $\mathbf{P}_{-r}(Z \in [t_{s-1}, t_s), Z' \in [t_{u-1}, t_u))$ with $(Z, Z')_r$ as in (1). It is shown in [28] that the MLE with $b \geq 2$ can be more efficient at the bit level than common single-bit quantization [10, 16]; the optimal choice of $b$ increases with $\rho$. While statistically optimal in the given setting, the MLE remains computationally cumbersome even when using the approximation in [28] because it requires cross-tabulation of the empirical frequencies corresponding to the bivariate normal probabilities above. This makes the use of the MLE unattractive particularly in situations in which it is not feasible to materialize all $O(n^2)$ pairwise similarities estimable from $(q_i, q_j)_{i<j}$ so that they would need to be re-evaluated frequently.

**Collision-based estimator.** The collision-based estimator proposed in [27] is a bin-based approach as the MLE. The similarity $\rho$ is estimated as $\widehat{\rho}_{\text{col}} = \theta^{-1}\left(\sum_{l=1}^k I(q_{(l)} = q'_{(l)})/k\right)$, where the map $\theta : [0,1] \to [0,1]$ is defined by $r \mapsto \theta(r) = \mathbf{P}_r(Q(Z) = Q(Z'))$, shown to be monotonically increasing in [27]. Compared to the MLE, $\widehat{\rho}_{\text{col}}$ uses less information – it only counts "collisions", i.e., events $\{q_{(l)} = q'_{(l)}\}$. The loss in statistical efficiency is moderate for $b = 2$, in particular for $\rho$ close to 1. However, as $b$ increases that loss becomes more and more substantial; cf. Figure 1. On the positive side, $\widehat{\rho}_{\text{col}}$ is convenient to compute given that the evaluation of the function $\theta^{-1}$ can be approximated by employing a look-up table after tabulating $\theta$ on a fine grid.

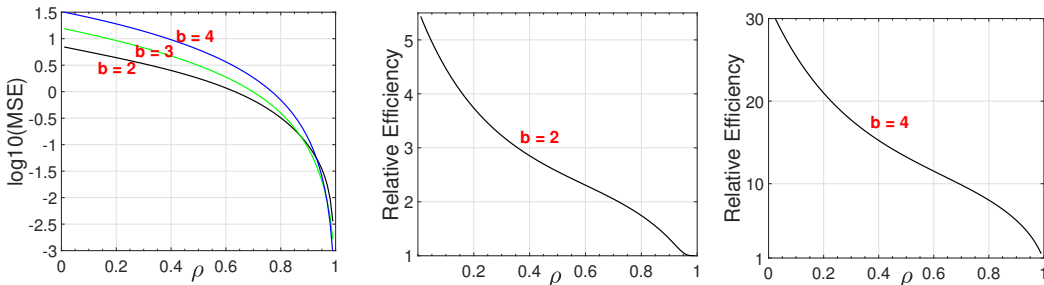

Figure 1: (L): Asymptotic MSEs [27] of $\widehat{\rho}_{\text{col}}$ (to be divided by $k$) for $2 \leq b \leq 4$. (M,R): Asymptotic relative efficiencies $\text{MSE}_\rho(\widehat{\rho}_{\text{col}})/\text{MSE}_\rho(\widehat{\rho}_{\text{MLE}})$ for $b \in \{2, 4\}$, where $\widehat{\rho}_{\text{MLE}}$ is the MLE in [28].

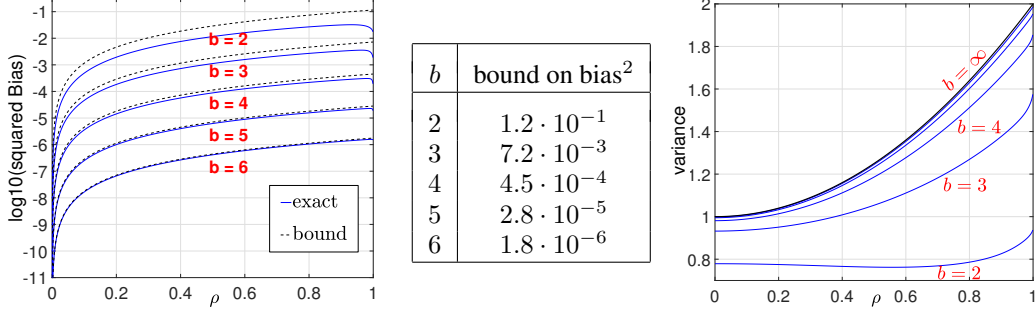

Figure 2: (L): $\mathrm{Bias}_\rho^2(\widehat{\rho}_{\mathrm{lin}})$ and the bound of Theorem 1. (M): uniform upper bounds on $\mathrm{Bias}_\rho^2(\widehat{\rho}_{\mathrm{lin}})$ obtained from Theorem 1 by setting $\rho = 1$. (R): $\mathrm{Var}_\rho(\widehat{\rho}_{\mathrm{lin}})$ (to be divided by $k$).

## 3.2 Code-based approaches

In the code-based approach, we simply ignore the fact that the quantized data actually represent intervals and treat them precisely in the same way as full-precision data. Recovery of cosine similarity is performed by means of the estimator in §2 with $z, z'$ replaced by $q, q'$. Perhaps surprisingly, it turns out that depending on $\rho$ the loss of information incurred by this rather crude approach can be small already for bit depths between $b = 3$ and $b = 5$. That loss increases with $\rho$, with a fundamental gap compared to bin-based approaches and to the full precision case in the limit $\rho \to 1$.

**Linear estimator.** We first consider $\widehat{\rho}_{\mathrm{lin}} = \langle q, q' \rangle / k$. We note that $\widehat{\rho}_{\mathrm{lin}} = \widehat{\rho}_{\mathrm{lin},b}$ depends on $b$; $b = \infty$ corresponds to the estimator $\widehat{\rho}_{\mathrm{lin}} = \widehat{\rho}_{\mathrm{lin},\infty}$ in §2 denoted by the same symbol. A crucial difference between the code-based and the bin-based approaches discussed above is that the latter have vanishing asymptotic squared bias of the order $O(k^{-2})$ for any $b$ [27, 28]. This is not the case for code-based approaches whose bias needs to be analyzed carefully. The exact bias of $\widehat{\rho}_{\mathrm{lin}}$ in dependence of $\rho$ and $b$ can be evaluated exactly numerically. Numerical evaluations of bias and variance of estimators discussed in the present section only rely on the computation of coefficients $\theta_{\alpha,\beta}$ defined by

$$\theta_{\alpha,\beta} := \mathbf{E}_\rho[Q(Z)^\alpha Q(Z')^\beta] = \sum_{\sigma,\sigma' \in \{-1,1\}} \sum_{s,u=1}^K \sigma^\alpha (\sigma')^\beta \mu_s^\alpha \mu_u^\beta \, \mathbf{P}_\rho \left( Z \in \sigma(t_{s-1}, t_s), Z' \in \sigma'(t_{u-1}, t_u) \right),$$
(11)

where $\alpha, \beta$ are non-negative integers and $(Z, Z')$ are bivariate normal (1) with $r = \rho$. Specifically, we have $\mathbf{E}_\rho[\widehat{\rho}_{\mathrm{lin}}] = \theta_{1,1}$, $\mathrm{Var}_\rho(\widehat{\rho}_{\mathrm{lin}}) = (\theta_{2,2} - \theta_{1,1}^2)/k$. In addition to exact numerical evaluation, we provide a bound on the bias of $\widehat{\rho}_{\mathrm{lin}}$ which quantifies explicitly the rate of decay in dependence $b$.

**Theorem 1.** *We have* $\mathrm{Bias}_\rho^2(\widehat{\rho}_{lin}) \le 4\rho^2 D_b^2$, *where* $D_b = \frac{3^{3/2} 2\pi}{12} 2^{-2b} \approx 2.72 \cdot 2^{-2b}$.

As shown in Figure 2 (L), the bound on the squared bias in Theorem 1 constitutes a reasonable proxy of the exact squared bias. The rate of decay is $O(2^{-4b})$. Moreover, it can be verified numerically that the variance in the full precision case upper bounds the variance for finite $b$, i.e., $\mathrm{Var}_\rho(\widehat{\rho}_{\mathrm{lin},b}) \le \mathrm{Var}_\rho(\widehat{\rho}_{\mathrm{lin},\infty})$, $\rho \in [0,1)$. Combining bias and variance, we may conclude that depending on $k$, the MSE of $\widehat{\rho}_{\mathrm{lin}}$ based on coarsely quantized data does not tend to be far from what is achieved with full precision data. The following two examples illustrate this point.

(i) Suppose $k = 100$ and $b = 3$. With full precision, we have $\mathrm{MSE}_\rho(\widehat{\rho}_{\mathrm{lin},\infty}) = (1+\rho^2)/k \in [.01, .02]$. From Figure 2 (M) and the observation that $\mathrm{Var}_\rho(\widehat{\rho}_{\mathrm{lin},3}) \le \mathrm{Var}_\rho(\widehat{\rho}_{\mathrm{lin},\infty})$, we find that the MSE can go up by at most $7.2 \cdot 10^{-3}$, i.e., it can at most double relative to the full precision case.

(ii) Suppose $k = 1000$ and $b = 4$. With the same reasoning as in (i), the MSE under quantization can increase at most by a factor of $1.45$ as compared to full precision data.

Figure 3 shows that these numbers still tend to be conservative. In general, the difference of the MSEs for $b = \infty$ on the one hand and $b \in \{3, 4, 5\}$ on the other hand gets more pronounced for large values of the similarity $\rho$ and large values of $k$. This is attributed to the (squared) bias of $\widehat{\rho}_{\mathrm{lin}}$. In particular, it does not pay off to choose $k$ significantly larger than the order of the squared bias.

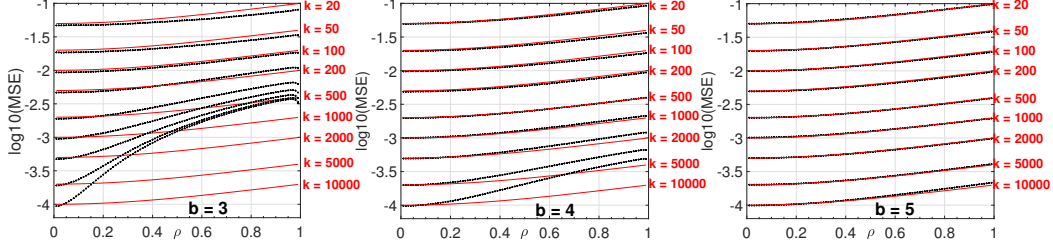

Figure 3: MSEs of $\widehat{\rho}_{\text{lin}}$ for various $k$ and $b \in \{3, 4, 5\}$ (dotted). The solid (red) lines indicate the corresponding MSEs for $\widehat{\rho}_{\text{lin}}$ in the full-precision case ($b = \infty$).

**Normalized estimator.** In the full precision case we have seen that simple normalization of the form $\widehat{\rho}_{\text{norm}} = \langle z, z' \rangle / (\|z\|_2 \|z'\|_2)$ can yield substantial benefits. Interestingly, it turns out that the counterpart $\widehat{\rho}_{\text{norm}} = \langle q, q' \rangle / (\|q\|_2 \|q'\|_2)$ for quantized data is even more valuable as it helps reducing the bias of $\widehat{\rho}_{\text{lin}} = \langle q, q' \rangle / k$. This effect can be seen easily in the limit $\rho \to 1$ in which case $\text{Bias}_\rho(\widehat{\rho}_{\text{norm}}) \to 0$ by construction. In general, bias and variance can be evaluated as follows.

**Proposition 1.** *In terms of the coefficients $\theta_{\alpha,\beta}$ defined in (11), as $k \to \infty$, we have*

$$|\operatorname{Bias}_\rho[\widehat{\rho}_{norm}]| = \left| \frac{\theta_{1,1}}{\theta_{2,0}} - \rho \right| + O(k^{-1})$$

$$\operatorname{Var}(\widehat{\rho}_{norm}) = \frac{1}{k}\left( \frac{\theta_{2,2}}{\theta_{2,0}^2} - \frac{2\theta_{1,1}\theta_{3,1}}{\theta_{2,0}^3} + \frac{\theta_{1,1}^2(\theta_{4,0}+\theta_{2,2})}{2\theta_{2,0}^4} \right) + O(k^{-2}).$$

Figure 4 (L,M) graphs the above two expressions. In particular, the plots highlight the reduction in bias compared to $\widehat{\rho}_{\text{lin}}$ and the fact that the variance is decreasing in $\rho$ as for $b = \infty$. While Proposition 1 is asymptotic, we verify a tight agreement in simulations for reasonably small $k$ (cf. supplement).

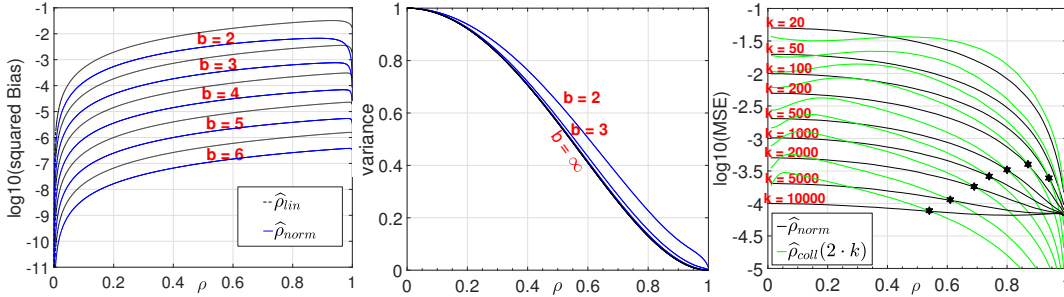

Figure 4: (L): Asymptotic $\operatorname{Bias}^2_\rho(\widehat{\rho}_{\text{norm}})$ relative to $\operatorname{Bias}^2_\rho(\widehat{\rho}_{\text{lin}})$. (M): $\operatorname{Var}_\rho(\widehat{\rho}_{\text{norm}})$ (asymptotic, to be divided by $k$). (R): MSEs of $\widehat{\rho}_{\text{lin},4}$ vs. the MSEs of $\widehat{\rho}_{\text{coll},2}$ using twice the number of RPs (comparison at the bit level). The stars indicate the values of $\rho$ at which the MSEs of the two estimators are equal.

### 3.3 Coding-based estimation vs. Collision-based estimation

Both schemes are comparable in terms of simplicity, but at the level of statistical performance none of the two dominates the other. The collision-based approach behaves favorably in a high similarity regime as shows a comparison of $\operatorname{MSE}_\rho(\widehat{\rho}_{\text{col}})$ ($b = 2$) and $\operatorname{MSE}_\rho(\widehat{\rho}_{\text{norm}})$ ($b = 4$) at the bit level (Figure 4 (R)): since $\widehat{\rho}_{\text{col}}$ uses only two bits for each of the $k$ RPs, while $\widehat{\rho}_{\text{norm}}$ uses twice as many bits, we have doubled the number of RPs for $\widehat{\rho}_{\text{col}}$. The values of $\rho$ for which the curves of the two approaches (for fixed $k$) intersect are indicated by stars. As $k$ decreases from $10^4$ to $10^2$, these values increase from about $\rho = 0.55$ to $\rho = 0.95$. In conclusion, $\widehat{\rho}_{\text{col}}$ is preferable in applications in which high similarities prevail, e.g., in duplicate detection. On the other hand, for generic high-dimensional data, one would rather not expect $\rho$ to take high values given that two points drawn uniformly at random from the sphere are close to orthogonal with high probability.

Figure 1 (L) shows that as $b$ is raised, $\widehat{\rho}_{\text{col}}$ requires $\rho$ to be increasingly closer to one to achieve lower MSE. By contrast, increasing $b$ for the coding-based schemes yields improvements essentially for the

whole range of $\rho$. An interesting phenomenon occurs in the limit $\rho \to 1$. It turns out that the rate of decay of $\mathrm{Var}_\rho(\widehat{\rho}_{\mathrm{norm}})$ is considerably slower than the rate of decay of $\mathrm{Var}_\rho(\widehat{\rho}_{\mathrm{col}})$.

**Theorem 2.** *For any finite $b$, we have*

$$\mathrm{Var}_\rho(\widehat{\rho}_{norm}) = \Theta((1-\rho)^{1/2}), \qquad \mathrm{Var}_\rho(\widehat{\rho}_{col}) = \Theta((1-\rho)^{3/2}) \quad as \ \rho \to 1.$$

The rate $\Theta((1-\rho)^{3/2})$ is the same as the MLE [28] which is slower than the rate $\Theta((1-\rho)^2)$ in the full precision case (cf. §2). We conjecture that the rate $\Theta((1-\rho)^{1/2})$ is intrinsic to code-based estimation as this rate is also obtained when computing the full precision MLE (3) with quantized data (i.e., $z, z'$ gets replaced by $q, q'$).

### 3.4 Quantization of norms

Let us recall that according to our basic setup in §1, we have assumed so far that it is possible to compute the norms $\lambda_i = \|x_i\|_2^2$, $i \in [n]$, of the original data prior to projection and quantization, and store them in full precision to approximately recover inner products and squared distances via

$$\langle x_i, x_j \rangle \approx \lambda_i \lambda_j \widehat{\rho}_{ij}, \qquad \|x_i - x_j\|_2^2 \approx \lambda_i^2 + \lambda_j^2 - 2\lambda_i \lambda_j \widehat{\rho}_{ij},$$

where $\widehat{\rho}_{ij}$ is an estimate of the cosine similarity of $x_i$ and $x_j$. Depending on the setting, it may be required to quantize the $\{\lambda_i\}_{i=1}^n$ as well. It turns out that the MSE for estimating distances can be tightly bounded in terms of the MSE for estimating cosine similarities and $\max_{1 \le i \le n} |\widehat{\lambda}_i - \lambda_i|$, where $\{\widehat{\lambda}_i\}_{i=1}^n$ denote the quantized versions of $\{\lambda_i\}_{i=1}^n$; the precise bound is stated in the supplement.

## 4 Empirical results: linear classification using quantized RPs

One traditional application of RPs is dimension reduction in linear regression or classification with high-dimensional predictors [14, 30]. The results of §3.2 suggest that as long as the number of RPs $k$ are no more than a few thousand, subsequent scalar quantization to four bits is not expected to have much of a negative effect relative to using full precision data. In this section, we verify this hypothesis for four high-dimensional data sets from the UCI repository: arcene ($d = 10^4$), Dexter ($d = 2 \cdot 10^4$), farm ($d = 5.5 \cdot 10^4$) and PEMS ($d = 1.4 \cdot 10^5$).

**Setup.** All data points are scaled to unit Euclidean norm before dimension reduction and scalar quantization based on the Lloyd-Max quantizer (10). The number of RPs $k$ is varied according to $\{2^6, 2^7, \ldots, 2^{12}\}$. For each of these values for $k$, we consider 20 independent realizations of the random projection matrix $A$. Given projected and quantized data $\{q_1, \ldots, q_n\}$, we estimate the underlying cosine similarities $\rho_{ij}$ as $\widehat{\rho}_{ij} = \widehat{\rho}(q_i, q_j)$, $i, j \in [n]$, where $\widehat{\rho}(q_i, q_j)$ is a placeholder for either the collision-based estimator $\widehat{\rho}_{\mathrm{coll}}$ based on $b = 2$ bits or the normalized estimator $\widehat{\rho}_{\mathrm{norm}}$ for $b \in \{1, 2, 4, \infty\}$ using data $\{q_{i_{(l)}}, q_{j_{(l)}}\}_{l=1}^k$; one-bit quantization ($b = 1$) is here included as a reference. The $\{\widehat{\rho}_{ij}\}_{1 \le i,j \le n}$ are then used as a kernel matrix fed into LIBSVM [9] to train a binary classifier. Prediction on test sets is performed accordingly. LIBSVM is run with 30 different values of its tuning parameter $C$ ranging from $10^{-3}$ to $10^4$.

**Results.** A subset of the results is depicted in Figure 5 which is composed of three columns (one for each type of plot) and four rows (one for each data set). All results are averages over 20 independent sets of random projections. The plots in the left column show the minimum test errors over all 30 choices of the tuning parameter $C$ under consideration in dependency of the number of RPs $k$. The plots in the middle column show the test errors in dependency of $C$ for a selected value of $k$ (the full set of plots can be found in the supplement). The plots in the right column provide a comparison of the minimum (w.r.t. $C$) test errors of $\widehat{\rho}_{\mathrm{coll},2}$ and $\widehat{\rho}_{\mathrm{norm},4}$ at the bit level, i.e., with $k$ doubled for $\widehat{\rho}_{\mathrm{coll},2}$. In all plots, classification performance improves as $b$ increases. What is more notable though is that the gap between $b = 4$ and $b = \infty$ is indeed minor as anticipated. Regarding the performance of $\widehat{\rho}_{\mathrm{coll},2}$ and $\widehat{\rho}_{\mathrm{norm},4}$, the latter consistently achieves better performance.

## 5 Conclusion

In this paper, we have presented theoretical and empirical evidence that it is possible to achieve additional data compression in the use of random projections by means of coarse scalar quantization.

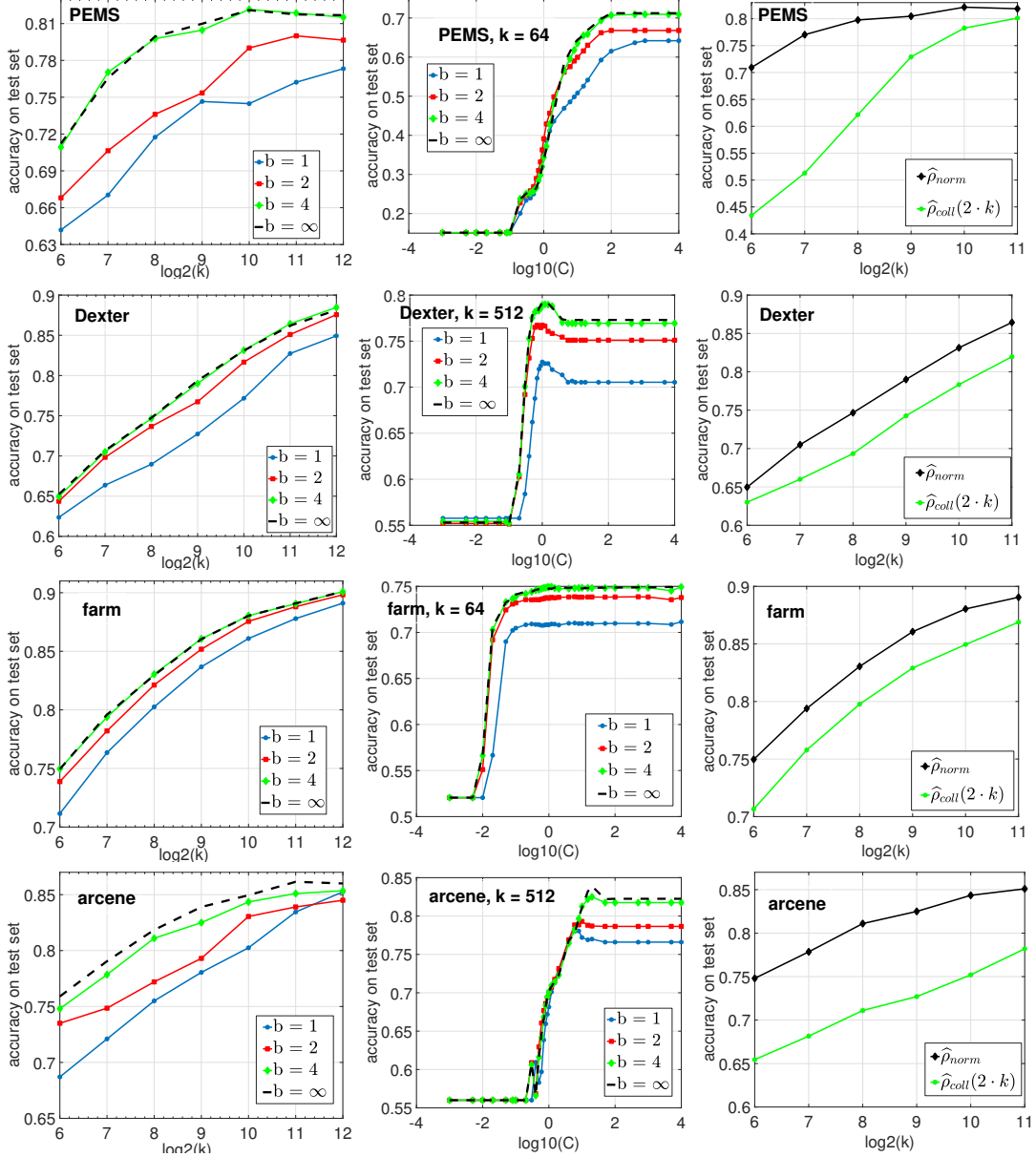

Figure 5: Results of the classification experiments. Each row corresponds to one data set. (L): Accuracy on the test set (optimized over $C$) in dependence of the number of RPs $k$ ($\log_2$ scale). (M): Accuracy on the test set for a selected value of $k$ in dependence of $\log_{10}(C)$. (R): Comparison of the test accuracies when using the estimators $\widehat{\rho}_{\text{norm},4}$ respectively $\widehat{\rho}_{\text{coll},2}$ with twice the number of RPs.

The loss of information incurred at this step tends to be mild even with the naive approach in which quantized data are treated in the same way as their full precision counterparts. An exception only arises for cosine similarities close to 1 (Theorem 2). We have also shown that the simple form of normalization employed in the construction of the estimator $\widehat{\rho}_{\text{norm}}$ can be extremely beneficial, even more so for coarsely quantized data because of a crucial bias reduction.

Regarding future work, it is worthwhile to consider the extension to the case in which the random projections are not Gaussian but arise from one of the various structured Johnson-Lindenstrauss transforms, e.g., those in [2, 3, 23]. A second direction of interest is to analyze the optimal trade-off between the number of RPs $k$ and the bit depth $b$ in dependence of the similarity $\rho$; in the present work, the choice of $b$ has been driven with the goal of roughly matching the full precision case.

**Acknowledgments**

The work was partially supported by NSF-Bigdata-1419210, NSF-III-1360971. Ping Li also thanks Michael Mitzenmacher for helpful discussions.

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
