[Supplementary Material · supplement-camera.pdf]

# Supplement to
# "Simple Strategies for Recovering Inner Products from Coarsely Quantized Random Projections"

**Ping Li**
Baidu Research, and
Rutgers University
pingli98@gmail.com

**Martin Slawski**
Department of Statistics
George Mason University
mslawsk3@gmu.edu

## Part I: Analysis

### A  Preparations

For the convenience of the reader, we here repeat material from the paper that will be frequently referred to in this supplement. For $r \in (-1, 1)$, we consider bivariate Gaussian random variables

$$(Z, Z')_r \sim N_2 \left( \begin{pmatrix} 0 \\ 0 \end{pmatrix}, \begin{pmatrix} 1 & r \\ r & 1 \end{pmatrix} \right). \tag{A.1}$$

For non-negative integers $\alpha, \beta$, we define coefficients $\theta_{\alpha,\beta}$ by

$$\theta_{\alpha,\beta} := \mathbf{E}_\rho[Q(Z)^\alpha Q(Z')^\beta]$$
$$= \sum_{\sigma,\sigma' \in \{-1,1\}} \sum_{s,u=1}^K \sigma^\alpha (\sigma')^\beta \mu_s^\alpha \mu_u^\beta \, \mathbf{P}_\rho \left( Z \in \sigma(t_{s-1}, t_s), Z' \in \sigma'(t_{u-1}, t_u) \right), \tag{A.2}$$

where $(Z, Z')$ are bivariate normal (A.1) with $r = \rho$. We recall that $\{t_k\}_{k=1}^{K-1}$ and $\{\mu_k\}_{k=1}^K$, $K = 2^{b-1}$, denote the tresholds and codes, respectively, assciated with a $b$-bit scalar quantizer $Q$.

### B  Proof of Theorem 1

Consider the linear estimator $\widehat{\rho}_{\text{lin}} = \langle q, q' \rangle / k$ based on quantized data $q, q'$.

**Theorem 1.** *We have* $\text{Bias}_\rho^2(\widehat{\rho}_{lin}) \le 4\rho^2 D_b^2$, *where* $D_b = \frac{3^{3/2} 2\pi}{12} 2^{-2b} \approx 2.72 \cdot 2^{-2b}$.

*Proof.* In the sequel, let $Q(\cdot) = Q_b(\cdot; \mathbf{t}^\star, \boldsymbol{\mu}^\star)$ be the Lloyd-Max quantizer at bit depth $b$ and let $D_b := \mathbf{E}[\{Z - Q(Z)\}^2]$ be the associated squared distortion. A standard property of the Lloyd-Max quantizer is that (cf. [2], p. 180)

$$\mathbf{E}[Q(Z)(Q(Z) - Z)] = 0. \tag{B.1}$$

This (B.1) immediately yields

$$\mathbf{E}[ZQ(Z)] = \mathbf{E}[Q(Z)^2] = \theta_{2,0}. \tag{B.2}$$

according to notation (A.2). Therefore,

$$D_b = \mathbf{E}[\{Z - Q(Z)\}^2] = \mathbf{E}[Z^2] - 2\mathbf{E}[ZQ(Z)] + \mathbf{E}[Q(Z)^2]$$
$$= \mathbf{E}[Z^2] - \mathbf{E}[Q(Z)^2] \tag{B.3}$$
$$= 1 - \theta_{2,0}.$$

Next, we note that

$$Z' \overset{\mathcal{D}}{=} \rho Z + \sqrt{1-\rho^2}\,\xi \qquad (B.4)$$

with $\xi \sim N(0,1)$ independent of $Z$, where $\overset{\mathcal{D}}{=}$ means equality in distribution. Combining (B.2) and (B.4), we obtain that

$$
\begin{aligned}
\mathbf{E}[Z'Q(Z)] &= \mathbf{E}[(\rho Z + \sqrt{1-\rho^2}\,\xi)Q(Z)] \\
&= \rho\,\mathbf{E}[ZQ(Z)] \\
&= \rho\theta_{2,0}.
\end{aligned}
\qquad (B.5)
$$

We now have

$$
\begin{aligned}
\mathbf{E}_\rho[\widehat{\rho}_{\mathrm{lin}}] = \mathbf{E}[Q(Z)Q(Z')] &= \mathbf{E}[(Z + \{Q(Z) - Z\})(Z' + \{Q(Z') - Z'\})] \\
&= \mathbf{E}[ZZ'] + \mathbf{E}[Z\{Q(Z') - Z'\}] + \mathbf{E}[Z'\{Q(Z) - Z\}] + \\
&\quad + \mathbf{E}[\{Q(Z) - Z\}\{Q(Z') - Z'\}] \\
&= \rho + 2\,\mathbf{E}[Z'\{Q(Z) - Z\}] + \mathbf{E}[\{Q(Z) - Z\}\{Q(Z') - Z'\}] \\
&= \rho + 2\rho(\theta_{2,0} - 1) + \mathbf{E}[\{Q(Z) - Z\}\{Q(Z') - Z'\}] \\
&= \rho - 2\rho D_b + \mathbf{E}[\{Q(Z) - Z\}\{Q(Z') - Z'\}],
\end{aligned}
\qquad (B.6)
$$

For the third identity from the top, we have used the fact that $Z$ and $Z'$ are exchangeable, and the last two identities follow from (B.5) and (B.3), respectively. Re-arranging (B.6), we obtain

$$\mathrm{Bias}_\rho^2(\widehat{\rho}_{\mathrm{lin}}) = (\mathbf{E}_\rho[\widehat{\rho}_{\mathrm{lin}}] - \rho)^2 = \{ -2\rho D_b + \mathbf{E}[\{Q(Z) - Z\}\{Q(Z') - Z'\}] \}^2 \qquad (B.7)$$

It is proved in Lemma 1 below that

$$0 \le \mathbf{E}[\{Q(Z) - Z\}\{Q(Z') - Z'\}] \le \rho D_b. \qquad (B.8)$$

Using that $\rho \ge 0$ and combining (B.7) and (B.8), it follows that

$$\mathrm{Bias}_\rho(\widehat{\rho}_{\mathrm{lin}})^2 \le 4\rho^2 D_b^2.$$

The following upper bound is well-known in the signal processing literature (cf. [3], p. 138)

$$D_b \le \frac{3^{3/2}2\pi}{12} \cdot 2^{-2b},$$

which yields the assertion.

$$\square$$

**Lemma 1.**

$$0 \le \mathbf{E}[\{Q(Z) - Z\}\{Q(Z') - Z'\}] \le \rho D_b. \qquad (B.9)$$

*Proof.* Define

$$
\begin{aligned}
\Delta_1(\rho) &= \mathbf{E}_\rho[ZZ'] - \mathbf{E}_\rho[ZQ(Z')] \\
\Delta_2(\rho) &= \mathbf{E}_\rho[ZQ(Z')] - \mathbf{E}_\rho[Q(Z)Q(Z')].
\end{aligned}
$$

In the sequel, we will establish that for all $\rho \in [0,1]$, it holds that

$$
\begin{aligned}
&(\mathsf{P1}) \quad \Delta_1(\rho) - \Delta_2(\rho) \ge 0, \\
&(\mathsf{P2}) \quad \Delta_2(\rho) \ge 0.
\end{aligned}
$$

Relations (P1) and (P2) immediately yield (B.9). Indeed, we have

$$
\begin{aligned}
\mathbf{E}_\rho[(Z - Q(Z))(Z' - Q(Z'))] &= (\mathbf{E}_\rho[ZZ'] - \mathbf{E}_\rho[ZQ(Z')]) - (\mathbf{E}_\rho[Z'Q(Z)] - \mathbf{E}_\rho[Q(Z)Q(Z')]) \\
&= \Delta_1(\rho) - \Delta_2(\rho) \ge 0,
\end{aligned}
$$

which yields the lower bound in (B.9). Likewise,

$$
\begin{aligned}
\mathbf{E}_\rho[(Z - Q(Z))(Z' - Q(Z'))] &= (\mathbf{E}_\rho[ZZ'] - \mathbf{E}_\rho[ZQ(Z')]) - (\mathbf{E}_\rho[Z'Q(Z)] - \mathbf{E}_\rho[Q(Z)Q(Z')]) \\
&= \Delta_1(\rho) - \Delta_2(\rho) \\
&\le \Delta_1(\rho) = \rho D_b,
\end{aligned}
$$

where the last identity follows with the same argument as used for (B.6) above. It thus remains to demonstrate (P1) and (P2).

Regarding (P2), we have

$$\Delta_2(0) = \mathbf{E}_{\rho=0}[ZQ(Z')] - \mathbf{E}_{\rho=0}[Q(Z)Q(Z')] = 0 \tag{B.10}$$

$$\Delta_2(1) = \mathbf{E}_{\rho=1}[ZQ(Z')] - \mathbf{E}_{\rho=1}[Q(Z)Q(Z')] = \mathbf{E}[ZQ(Z)] - \mathbf{E}[Q(Z)^2] = 0, \tag{B.11}$$

using (B.2) again. Property (P2) then follows from the following property:

(P3)   The map $\rho \mapsto \theta_{1,1}(\rho)$ is convex on $[0,1]$.

In fact, property (P3), which follows from Lemma 2 below, in turn implies that

$$\Delta_2(\rho) = \mathbf{E}_\rho[ZQ(Z')] - \mathbf{E}_\rho[Q(Z)Q(Z')] = \rho\theta_{2,0} - \theta_{1,1}(\rho),$$

is a concave function of $\rho$. Combining (B.10) and (B.11) with Jensen's inequality then yields that $\Delta_2(\rho) \geq 0$ for $\rho \in [0,1]$.

Regarding (P1), we expand

$$\begin{aligned}
\Delta_1(\rho) - \Delta_2(\rho) &= (\rho - \rho\theta_{2,0}) - (\rho\theta_{2,0} - \theta_{1,1}(\rho)) \\
&= \rho(1 - 2\theta_{2,0}) + \theta_{1,1}(\rho).
\end{aligned}$$

We first note that

$$\Delta_1(0) - \Delta_2(0) = 0 \tag{B.12}$$

To deduce (P1) given (B.12), one then shows that the map

$$\rho \mapsto \Delta_1(\rho) - \Delta_2(\rho) \tag{B.13}$$

is non-decreasing on $[0,1]$. We do this by obtaining the derivative

$$\gamma(\rho) := \frac{d}{d\rho}(\Delta_1(\rho) - \Delta_2(\rho)) = 1 - 2\theta_{2,0} + \frac{d}{d\rho}\theta_{1,1}(\rho).$$

Denoting $\nu(\rho) := \frac{d}{d\rho}\theta_{1,1}(\rho)$, one computes that

$$\nu(0) = \theta_{2,0}^2 \tag{B.14}$$

as an immediate consequence of Lemma 2 below. Substituting this back into $\gamma$, one obtains

$$\gamma(0) = 1 - 2\theta_{2,0} + \theta_{2,0}^2 = (1 - \theta_{2,0})^2 > 0.$$

Given (P3), $\frac{d}{d\rho}\theta_{1,1}(\rho)$ is non-decreasing on $[0,1]$ and hence $\gamma(\rho) \geq 0$ on $[0,1]$, which in turn implies that the map (B.13) is non-decreasing. Combining this with (B.12) yields (P1) and thus the proof of the lemma. $\qquad \square$

Property (P3) and (B.14) can be deduced from the following lemma.

**Lemma 2.** *Let $f : \mathbb{R} \to \mathbb{R}$ be uniformly bounded. Consider the following map defined on $[0,1]$:*

$$\rho \mapsto \eta_f(\rho) := \mathbf{E}_\rho[f(Z)f(Z')], \qquad (Z, Z') \sim N_2\left(\begin{bmatrix} 0 \\ 0 \end{bmatrix}, \begin{bmatrix} 1 & \rho \\ \rho & 1 \end{bmatrix}\right).$$

*Then $\eta_f$ obeys the following series expansion:*

$$\eta_f(\rho) = \sum_{k=0}^{\infty} \frac{\rho^k}{k!}\left(\int f(x)H_k(x)\phi(x)\,dx\right)^2, \tag{B.15}$$

*where $\phi$ denotes the standard Gaussian PDF and $H_k$ is the $k$-th Hermite polynomial defined by*

$$H_k(x) = (-1)^k \exp(x^2/2)\frac{d^k}{dx^k}\exp(-x^2/2), \quad k = 0, 1, \ldots$$

*Proof.* From a result in [1], p. 133, the PDF of $(Z, Z')$, say $\phi_\rho$, can be expanded as

$$\phi_\rho(x, y) = \sum_{k=0}^{\infty} \frac{\rho^k}{k!} H_k(x) H_k(y) \phi(x) \phi(y).$$

Using this result, we obtain that

$$\eta_f(\rho) = \int \int f(x) f(y) \, \phi_\rho(x, y) \, dx \, dy$$

$$= \int \int f(x) f(y) \sum_{k=0}^{\infty} \frac{\rho^k}{k!} H_k(x) H_k(y) \phi(x) \phi(y) dx \, dy$$

Since $f$ is uniformly bounded, the $\{H_k\}_{k=0}^{\infty}$ are polynomials and $\phi$ is a Schwartz function, each partial sum associated with the series inside the integrand is uniformly bounded. We may hence appeal to the bounded convergence theorem to obtain that

$$\eta_f(\rho) = \sum_{k=0}^{\infty} \frac{\rho^k}{k!} \int \int f(x) f(y) H_k(x) H_k(y) \phi(x) \phi(y) dx \, dy$$

$$= \sum_{k=0}^{\infty} \frac{\rho^k}{k!} \left( \int f(x) H_k(x) \phi(x) \, dx \right)^2.$$

$\square$

We apply Lemma 2 with the quantization map, i.e., $f = Q$. Expansion (B.15) yields that the second derivative of $\theta_{1,1}(\rho)$ is non-negative, and thus convexity. Similarly, (B.14) can be obtained by term-wise differentiation of (B.15), and evaluation of the result at zero, noting that $H_1(x) = x$, and using again that $\mathbf{E}[ZQ(Z)] = \theta_{2,0}$ (B.2).

## C  Proof of Proposition 1

Consider the normalized estimator $\widehat{\rho}_{\text{lin}} = \langle q, q' \rangle / (\|q\|_2 \|q'\|_2)$ based on quantized data $q, q'$.

**Proposition 1.** *In terms of the coefficients $\theta_{\alpha,\beta}$ defined in (A.2), as $k \to \infty$, we have*

$$|\operatorname{Bias}_\rho[\widehat{\rho}_{norm}]| = \left| \frac{\theta_{1,1}}{\theta_{2,0}} - \rho \right| + O(k^{-1}), \tag{C.1}$$

$$\operatorname{Var}(\widehat{\rho}_{norm}) = \frac{1}{k} \left( \frac{\theta_{2,2}}{\theta_{2,0}^2} - \frac{2\theta_{1,1}\theta_{3,1}}{\theta_{2,0}^3} + \frac{\theta_{1,1}^2(\theta_{4,0} + \theta_{2,2})}{2\theta_{2,0}^4} \right) + O(k^{-2}). \tag{C.2}$$

*Proof.* We first show (C.1). From a first-order Taylor expansion of $(a, b) \mapsto \frac{a}{b}$ around $(a_0, b_0)$, we have

$$\frac{a}{b} = \frac{a_0}{b_0} + \frac{(a - a_0)}{b_0} - \frac{(b - b_0)a_0}{b_0^2} + O((a - a_0)^2) + O((b - b_0)^2) \quad \text{as } a \to a_0, \ b \to b_0.$$

Using this with $a = \langle q, q' \rangle / k$ and $b = \sqrt{\frac{1}{k}\|q\|_2^2 \frac{1}{k}\|q'\|_2^2}$, $a_0 = \mathbf{E}[a]$, $b_0 = \mathbf{E}[b]$, and taking expectations, we obtain that

$$\mathbf{E}[\widehat{\rho}_{\text{norm}}] = \mathbf{E}\left[ \frac{\frac{1}{k}\langle q, q' \rangle}{\sqrt{\frac{1}{k}\|q\|_2^2 \frac{1}{k}\|q'\|_2^2}} \right] = \frac{\mathbf{E}[\widehat{\rho}_{\text{lin}}]}{\mathbf{E}\left[\sqrt{\frac{1}{k}\|q\|_2^2 \frac{1}{k}\|q'\|_2^2}\right]} + O(1/k) \quad \text{as } k \to \infty. \tag{C.3}$$

Let us now turn our attention to the expectation in the denominator. Let $X_0 = \frac{1}{k}\|q\|_2^2 \frac{1}{k}\|q'\|_2^2$ and $E_0 = \mathbf{E}[X_0]$. We have

$$\mathbf{E}\left[\sqrt{\frac{1}{k}\|q\|_2^2 \frac{1}{k}\|q'\|_2^2}\right] = \mathbf{E}\left[\sqrt{E_0} + \frac{X_0 - E_0}{2\sqrt{E_0}} + O((X_0 - E_0)^2)\right] \quad \text{as } X_0 \to E_0 \tag{C.4}$$

$$= \sqrt{E_0} + O(1/k) \quad \text{as } k \to \infty. \tag{C.5}$$

For $E_0$ we obtain that

$$
\begin{aligned}
E_0 &= \mathbf{E}\left[\frac{1}{k}\|q\|_2^2 \, \frac{1}{k}\|q'\|_2^2\right] \\
&= \mathbf{E}\left[\left\{\frac{1}{k}\sum_{l=1}^{k}Q(z_{(l)})^2\right\}\left\{\frac{1}{k}\sum_{m=1}^{k}Q(z'_{(m)})^2\right\}\right] \\
&= \mathbf{E}\left[\frac{1}{k^2}\sum_{l\neq m}Q(z_{(l)})^2 Q(z'_{(m)})^2\right] + \mathbf{E}\left[\frac{1}{k^2}\sum_{l=1}^{k}Q(z_{(l)})^2 Q(z'_{(l)})^2\right] \\
&= \frac{k(k-1)}{k^2}\mathbf{E}[Q(z_{(1)})]^2 + \frac{1}{k}\mathbf{E}[Q(z_{(1)})^2 Q(z'_{(1)})^2] = \Psi^4 + O(1/k) \text{ as } k \to \infty. \qquad \text{(C.6)}
\end{aligned}
$$

For the last line, we use that for $l \neq m$, $z_{(l)}$ and $z'_{(m)}$ are independent and that $\{Q(z_{(l)}), Q(z'_{(l)})\}_{l=1}^{k}$ are identically distributed. Combining (C.3), (C.4), (C.6) and using one more first-order Taylor expansion for the resulting $O(1/k)$ term in the denominator in (C.3), the result (C.1) follows.

Let us now turn to the expression for the variance (C.2). Let $a = \langle q, q'\rangle/k$, $b = \|q\|_2^2/k$, and $c = \|q'\|_2^2/k$ and consider the function $\phi(a, b, c) = a/\sqrt{b \cdot c}$ so that $\mathrm{Var}(\widehat{\rho}_{\text{norm}}) = \mathrm{Var}(\phi(a, b, c))$. By a first-order Taylor expansion of $\phi$ around $(\mathbf{E}[a], \mathbf{E}[b], \mathbf{E}[c])$, we obtain

$$
\mathrm{Var}(\widehat{\rho}_{\text{norm}}) = \nabla\phi(\mathbf{E}[a], \mathbf{E}[b], \mathbf{E}[c])^\top \mathrm{Cov}_{a,b,c}\nabla\phi(\mathbf{E}[a], \mathbf{E}[b], \mathbf{E}[c]) + O(1/k^2), \text{ as } k \to \infty,
$$

where

$$
\mathrm{Cov}_{a,b,c} = \frac{1}{k}\begin{pmatrix} \theta_{2,2} - \theta_{1,1}^2 & \theta_{3,1} - \theta_{1,1}\theta_{2,0} & \theta_{3,1} - \theta_{1,1}\theta_{2,0} \\ \theta_{3,1} - \theta_{1,1}\theta_{2,0} & \theta_{4,0} - \theta_{2,0}^2 & \theta_{2,2} - \theta_{2,0}^2 \\ \theta_{3,1} - \theta_{1,1}\theta_{2,0} & \theta_{2,2} - \theta_{2,0}^2 & \theta_{4,0} - \theta_{2,0}^2 \end{pmatrix}, \qquad \text{(C.7)}
$$

$$
\nabla\phi(a, b, c) = \left(\frac{1}{\sqrt{b \cdot c}}, -\frac{1}{2}b^{-3/2}\frac{a}{\sqrt{c}}, -\frac{1}{2}c^{-3/2}\frac{a}{\sqrt{b}}\right)^\top.
$$

The expression for the covariance $\mathrm{Cov}_{a,b,c}$ of $a$, $b$ and $c$ is obtained by direct calculation in terms of coefficients (A.2). Evaluating $\nabla\phi(a, b, c)$ at $(\mathbf{E}[a] = \theta_{1,1}, \mathbf{E}[b] = \theta_{2,0}, \mathbf{E}[c] = \theta_{2,0})$ yields

$$
\nabla\phi(\mathbf{E}[a], \mathbf{E}[b], \mathbf{E}[c]) = \left(\frac{1}{\theta_{2,0}}, -\frac{\theta_{1,1}}{2\theta_{2,0}^2}, -\frac{\theta_{1,1}}{2\theta_{2,0}^2}\right)^\top.
$$

The final expression (C.2) results by expanding the quadratic form and collecting terms. $\qquad \square$

## D   Proof of Theorem 2

**Theorem 2.** *For any finite b, we have*

$$
\mathrm{Var}_\rho(\widehat{\rho}_{norm}) = \Theta((1-\rho)^{1/2}), \qquad \mathrm{Var}_\rho(\widehat{\rho}_{col}) = \Theta((1-\rho)^{3/2}) \quad \text{as } \rho \to 1.
$$

*Sketch.*

According to Proposition 1, we have that

$$
\mathrm{Var}_\rho(\widehat{\rho}_{\text{norm}}) = \frac{1}{k}\left(\frac{\theta_{2,2}}{\theta_{2,0}^2} - \frac{2\theta_{1,1}\theta_{3,1}}{\theta_{2,0}^3} + \frac{1}{2}\frac{\theta_{1,1}^2(\theta_{4,0} + \theta_{2,2})}{\theta_{2,0}^4}\right) + O(1/k^2) \text{ as } k \to \infty. \qquad \text{(D.1)}
$$

We have $\theta_{1,1} \to \theta_{2,0}$, as well as $\theta_{3,1} \to \theta_{4,0}$ and $\theta_{2,2} \to \theta_{4,0}$ as $\rho \to 1$. Based on arguments made in the proof of Theorem 1 in [4], it can be verified that the rate of convergence for all these limits is $\Theta(\sqrt{1-\rho})$. Expanding the fraction in (D.1), it can then be seen that the numerator converges to zero at rate $\Theta(\sqrt{1-\rho})$, while the denominator $\theta_{2,0}^4$ does not depend on $\rho$.

The rate of decay of $\mathrm{Var}_\rho(\widehat{\rho}_{\text{col}})$ can be directly deduced from Theorem 1 in [4], since in the limit the collision-based estimator $\widehat{\rho}_{\text{col}}$ coincides with the maximum likelihood estimator whose variance has been shown to decay at the rate $\Theta((1-\rho)^{3/2})$.

# E  Quantization of norms (§3.4 in the paper)

Let $x, x'$ be a generic set of points from $\mathcal{X} = \{x_1, \ldots, x_n\}$, and let $\lambda = \|x\|_2$ and $\lambda' = \|x'\|_2$ denote their norms. After quantizing the norms, we obtain $\widehat{\lambda}$ instead of $\lambda$ and $\widehat{\lambda}'$ instead of $\lambda'$. Let $\widehat{\rho}$ be an estimator of the cosine similarity $\rho = \frac{\langle x, x' \rangle}{\lambda \cdot \lambda'}$ of $x$ and $x'$, and consider the following estimator of the squared distance $\boldsymbol{d} = \|x - x'\|_2^2$:

$$\widehat{\boldsymbol{d}^2} = \widehat{\lambda}^2 + \widehat{\lambda}'^2 - 2\widehat{\lambda}\widehat{\lambda}'\widehat{\rho}$$

The MSE of $\widehat{\boldsymbol{d}^2}$ can then be bounded in terms of the MSE of $\widehat{\rho}$ and $\varepsilon = \max\{|\widehat{\lambda} - \lambda|, |\widehat{\lambda}' - \lambda'|\}$.

**Proposition 2.**

$$\mathbf{E}_\rho[\{\widehat{\boldsymbol{d}^2} - \boldsymbol{d}^2\}^2] \le 4\lambda^2(\lambda')^2\, \mathbf{E}_\rho[\{\widehat{\rho} - \rho\}^2] + 8\lambda\lambda'(\lambda + \lambda')\varepsilon\,(2|\operatorname{Bias}_\rho(\widehat{\rho})| + \operatorname{Var}_\rho(\widehat{\rho})) + O(\varepsilon^2). \quad \text{(E.1)}$$

*Proof.* Let us denote $\delta = \widehat{\lambda} - \lambda$ and $\delta' = \widehat{\lambda}' - \lambda'$. We then have

$$\widehat{\boldsymbol{d}^2} = \widehat{\lambda}^2 + \widehat{\lambda}'^2 - 2\widehat{\lambda}\widehat{\lambda}'\widehat{\rho} = \lambda^2 + 2\delta\lambda + \lambda'^2 + 2\delta'\lambda' - 2\lambda\lambda'\widehat{\rho} - 2(\lambda\delta' + \lambda'\delta)\widehat{\rho} + O(\varepsilon^2)$$

and thus

$$\widehat{\boldsymbol{d}^2} - \boldsymbol{d}^2 = \underbrace{2\lambda\lambda'(\rho - \widehat{\rho})}_{L} + \underbrace{2\lambda(\delta - \delta'\widehat{\rho})}_{R_1} + \underbrace{2\lambda'(\delta' - \delta\widehat{\rho})}_{R_2} + O(\varepsilon^2).$$

Define $R = R_1 + R_2$. Then

$$\begin{aligned}
\mathbf{E}_\rho[\{\widehat{\boldsymbol{d}^2} - \boldsymbol{d}^2\}^2] &= \mathbf{E}_\rho[(L+R)^2] = \mathbf{E}_\rho[L^2] + 2(\mathbf{E}_\rho[LR_1] + \mathbf{E}_\rho[LR_2]) + \mathbf{E}_\rho[R^2] + O(\varepsilon^2) \\
&= 4\lambda^2\lambda'^2\, \mathbf{E}_\rho[(\rho - \widehat{\rho})^2] + 2(\mathbf{E}_\rho[LR_1] + \mathbf{E}_\rho[LR_2]) + O(\varepsilon^2)
\end{aligned}$$
$$\text{(E.2)}$$

It remains to bound $\mathbf{E}_\rho[LR_1]$ and $\mathbf{E}_\rho[LR_2]$. By collecting terms, we obtain that

$$\begin{aligned}
\mathbf{E}_\rho[LR_1] &= 4\lambda^2\lambda'\, \mathbf{E}_\rho[(\rho - \widehat{\rho})(\delta - \delta'\widehat{\rho})] \\
&= 4\lambda^2\lambda'\left\{\delta(\rho - \mathbf{E}_\rho[\widehat{\rho}]) - \delta'\rho\,\mathbf{E}_\rho[\widehat{\rho}] + \delta'\,\mathbf{E}_\rho[\widehat{\rho}^2]\right\} \\
&= 4\lambda^2\lambda'\left\{\delta(\rho - \mathbf{E}_\rho[\widehat{\rho}]) - \delta'\rho\,\mathbf{E}_\rho[\widehat{\rho}] + \delta'(\operatorname{Var}_\rho(\widehat{\rho}) + \mathbf{E}_\rho[\widehat{\rho}]^2)\right\} \\
&= 4\lambda^2\lambda'\left\{\delta(\rho - \mathbf{E}_\rho[\widehat{\rho}]) + \delta'\,\mathbf{E}_\rho[\widehat{\rho}](\mathbf{E}_\rho[\widehat{\rho}] - \rho) + \delta'\operatorname{Var}_\rho(\widehat{\rho})\right\} \\
&\le 4\lambda^2\lambda'\varepsilon(2|\operatorname{Bias}_\rho(\widehat{\rho})| + \operatorname{Var}_\rho(\widehat{\rho}))
\end{aligned}$$
$$\text{(E.3)}$$

Similarly, it can be shown that

$$\mathbf{E}_\rho[LR_2] \le 4\lambda\lambda'^2\varepsilon(2|\operatorname{Bias}_\rho(\widehat{\rho})| + \operatorname{Var}_\rho(\widehat{\rho})) \qquad \text{(E.4)}$$

Combining (E.2), (E.3) and (E.4), we conclude the result. $\qquad\square$

# Part II: Additional Figures

## Empirical verification of the asymptotic expressions in Proposition 1

The plots compares the asymptotic MSE of $\widehat{\rho}_{\text{norm}}$ according to Proposition 1 (solid grey line) to the empirical MSEs (black dots) for $\rho \in \{0.01, 0.02, \ldots, 0.99\}$ based on $10^4$ independent simulations for different choices of $k$.

**Full set of plots for §4**

farm, k = 64

farm, k = 128

farm, k = 256

farm, k = 512

farm, k = 1024

farm, k = 2048

farm, k = 4096

accuracy on test set

log10(C)

b = 1
b = 2
b = 4
b = ∞