[Reviews · NeurIPS 2017]

Reviewer 1



Paper Summary: The authors study the distortion induced on inner products from applying a quantization over random projection. They show that even a highly coarse quantization (of 3-5 bits) only results in negligible to distortion overall (but significantly more distortion is induced on pairs of points with high cosine similarity). Review: I am not too familiar with the quantization after random projection literature to comment on the originality or significance of this work. Overall I like the approach to compress the data even further to gain significant speedups without compromising on quality. I believe that the work presented here can help practitioners gain better understanding on when and how simple quantization can help as a post processing step after random projection.

Reviewer 2



******************************** * Summary * ******************************** The paper investigates theoretically and empirically different strategies for recovery of inner products using quantized random projections of data instances. Random projections are often used in learning tasks involving dimensionality reduction. The goal of the additional quantization step is data compression that allows for a reduction in space complexity of learning algorithms and more efficient communication in distributed settings. ******************************** * Theoretical contributions * ******************************** The main focus of the paper is on studying a linear strategy for recovery of inner products from quantized random projections of the data. The strategy approximates the inner product between two instances from the instance space with the inner product of the corresponding quantized random projections divided by the dimension of the projection space. The main theoretical contribution is a bound on the bias of such approximations (Theorem 1). In addition to this strategy, the paper considers recovery of inner products from random projections that are normalized (i.e., having unit norm) prior to quantization. For such approximations, the paper expresses the bias and variance in terms of the relevant terms of the linear strategy for recovery of inner products (Proposition 1). The paper also provides a bound on the variance of recovery of inner products with values close to one and strategies based on quantization with finitely many bits (Theorem 2). ******************************** * Quantization * ******************************** The quantization of random projections is performed using the Lloyd-Max quantizer. The method resembles one dimensional K-means clustering where interval end-points determine the clusters and the centroid is given as the conditional expectation of the standard normal random variable given the interval, i.e., c_k = E [ z | z \in [t_k, t_{k+1}) ], where c_k is the cluster centroid or quantization value, t_k and t_{k + 1} are interval end-points of the kth interval, and the total number of intervals K is given with the number of quantization bits B = 1 + \log_2 K. ******************************** * Empirical study * ******************************** # Figure 2 The goal of this experiment is to numerically verify the tightness of the bound on the bias of the linear strategy for recovery of inner products from quantized random projections. The figure shows that the bound is tight and already for the recovery with 5-bit quantization the bound almost exactly matches the bias. The paper also hypothesizes that the variance of the linear strategy with finite bit-quantization is upper bounded by the variance of the same strategy without quantization. The provided empirical result is in line with the hypothesis. # Figure 3 The experiment evaluates the mean squared error of the recovery of inner products using the linear strategy as the number of quantization bits and dimension of the projection space change. The plot indicates that quantization with four bits and a few thousand of random projections might suffice for a satisfactory recovery of inner products. # Figure 4 - In the first experiment, the normalized strategy for recovery of inner products from quantized random projections is compared to the linear one. The plot (left) indicates that a better bias can be obtained using the normalized strategy. - In the second experiment, the variance of the quantized normalized strategy is compared to that without quantization. The plot (middle) indicates that already for quantization with 3 bits the variance is very close to the asymptotic case (i.e., infinitely many bits and no quantization). - The third experiment compares the mean squared error of the normalized strategy for recovery of inner products from quantized random projections to that of the collision strategy. While the collision strategy performs better for recovery of inner products with values close to one, the normalized strategy is better globally. # Figure 5 The experiment evaluates the strategies for recovery of inner products on classification tasks. In the first step the random projections are quantized and inner products are approximated giving rise to a kernel matrix. The kernel matrix is then passed to LIBSVM that trains a classifier. The provided empirical results show that the quantization with four bits is capable of generating an approximation to kernel matrix for which the classification accuracy matches that obtained using random projections without quantization. The plots depict the influence of the number of projections and the SVM hyperparameter on the accuracy for several high-dimensional datasets. The third column of plots in this figure also demonstrates that the normalized strategy for recovery of inner products from quantized random projections is better on classification tasks than the competing collision strategy. ******************************** * Theorem 1 * ******************************** Please correct me if I misunderstood parts of the proof. # Appendix B: Eq. (4) --> Bound Combining Eq. (6) with Eq. (4) it follows that E[\rho_{lin}] - \rho = -2 \rho D_b + E[(Q(Z) - Z)(Q(Z') - Z')] >= -2 \rho D_b ==> 2 \rho D_b >= \rho - E[\rho_{lin}] . To be able to square the latter without changing the inequality one needs to establish that \rho - E[\rho_{lin}] >= 0. Otherwise, it is possible that | \rho - E[\rho_{lin}] | > 2 \rho D_b and \rho - E[\rho_{lin}] < 0. # Appendix B: Eq. (6) - A proof of the left-hand side inequality is incomplete, E[(Q(Z) - Z)(Q(Z') - Z')] >= 0. At the moment the term is just expanded and it is claimed that the expansion is a fact. If so, please provide a reference for this result. Otherwise, an explanation is needed for why it holds that E[ Z Q(Z') ] - E[ Q(Z) Q(Z') ] <= E[ Z Z'] - E[ Z Q(Z') ]. - For the proof of the right-hand side inequality, it is not clear why it holds that E[ Z Z' ] + E[ Q(Z) Q(Z') ] - 2 E[ Z Q(Z') ] <= E[ Z Z' ] - E[ Z Q(Z') ]. ### My decision is conditional on these remarks being addressed properly during the rebuttal phase. ### ******************************** * Minor comments * ******************************** - line 79: bracket is missing ( || z ||^2 + || z' ||^2 ) / k - Appendix A, Eq. (2): the notation is not introduced properly

Reviewer 3



In this paper, authors investigate the dimensionality reduction technique with random projection followed by subsequent coarse quantization, which compresses data further. They find that with only 3-5 bits per each projection, the loss in accuracy becomes generally acceptable for classification tasks. Their approach does not require complicated steps such as maximum likelihood estimation that is required by previous work. The proposed normalized linear estimator comes with a variance at most 2x that of previous MLE based method. And the analysis of quantization clearly shows that just a few bits would reduce the variance asymptotically towards that with infinite precision. Generally speaking, it would be very interesting if authors can apply some data dependent feature learning techniques such as autoencoders and compare the performance under same bit length.